# Monitoring Shear-Zone Weakening in East Antarctic Outlet Glaciers through Differential InSAR Measurements

Christian T. Wild<sup>1,8</sup>, Reinhard Drews<sup>1</sup>, Niklas Neckel<sup>2</sup>, Joohan Lee<sup>3</sup>, Sihyung Kim<sup>3</sup>, Hyangsun Han<sup>4</sup>, Won Sang Lee<sup>5</sup>, Veit Helm<sup>2</sup>, Sebastian Harry Reid Rosier<sup>6</sup>, Oliver J. Marsh<sup>7,8</sup> and Wolfgang Rack<sup>8</sup>

<sup>1</sup>Department of Geosciences, University of Tübingen, Tübingen, Germany

<sup>2</sup>Alfred-Wegener-Institut Helmholtz-Zentrum für Polar- und Meeresforschung, Bremerhaven, Germany

<sup>3</sup>Center of Technology Development, Korea Polar Research Institute, Incheon, South Korea

<sup>4</sup>Department of Geophysics, Kangwon National University, South Korea

<sup>5</sup>Division of Glacier and Earth Sciences, Korea Polar Research Institute, Incheon, South Korea

<sup>6</sup>Department of Geography, University of Zürich, Zürich, Switzerland

<sup>7</sup>British Antarctic Survey, Cambridge, United Kingdom

5

<sup>8</sup>Gateway Antarctica, University of Canterbury, Christchurch, New Zealand

15 *Correspondence to*: Christian T. Wild (christian.wild@uni-tuebingen.de)

Abstract. The stability of the Antarctic Ice Sheet depends on ice flux into the ocean through major outlet glaciers, which is resisted by shear stresses in the lateral shear margins both on grounded ice and on floating ice shelves. Within the tidal flexure zone, where the ice sheet transitions from fully grounded to freely floating, ocean tides lead to a characteristic flexural pattern which can be detected by radar satellites in differential interferograms. Here, we investigate how spatially heterogeneous, elastic ice-shelf properties in the shear zones affect tidal flexure and if a corresponding signature can be detected in satellite observations. We use the Young's modulus (which among others depends on ice temperature and/or ice crystal orientation fabric and damage) as a bulk tuning variable for changing ice stiffness across shear zones and show that this leads to cm-scale deviations in vertical displacement compared to a homogeneous elastic flexure model. Using the tidal-flexure zone of Priestley Glacier as an example, we compare homogenous and heterogenous flexure-model predictions with observations from 31 differential interferograms. After adjusting the local tide model and validating it with in-situ GPS data, we find that a five-fold reduction of the Young's modulus in the shear zone, i.e. an effective shear-zone weakening, reduces the root-mean-square-error of predicted and observed vertical displacement by 33 % within the central part of the ice shelf. This suggests that satellite interferometry can detect changing ice stiffness across shear zones with potential to inform ice-flow models about the often unknown spatial variability in ice-shelf properties along the grounding zone.

# 1 Introduction

30

# 1.1 Context and research question

Outlet glaciers transport ice from the interior of the Antarctic ice sheet to the coast where ice crosses the grounding line and forms floating ice shelves. Ice shelves are largely in hydrostatic equilibrium with the ocean, apart from a few kilometer wide belt just seawards of the grounding line, where the ice continuously flexes between the fully grounded and the fully floating areas. The pattern of the vertical displacement arising from this flexure has stirred much previous research for example because it modulates the dynamics of tributary ice streams (Anandakrishnan et al., 2003; Gudmundsson, 2006; Murray et al., 2007), because it may foster exchange of subglacial and ocean water (Walker et al., 2013; Warburton et al., 2020; Robel et al., 2022; Bradley and Hewitt, 2024), because it reveals tidal grounding-line migration (Freer et al., 2023), and also because the ice here is not in hydrostatic equilibrium and so confidently deriving ice thickness at the grounding line from satellite altimetry is a significant challenge (Fricker and Padman, 2006; Fricker et al., 2009). Here, we explore if the vertical displacement by tides can be used as a natural experiment to highlight spatially variable ice-shelf properties resulting in spatially variable ice stiffness. For example, if the ice stiffness varies spatially and is lower in the lateral shear margins than elsewhere, this would reduce ice-shelf buttressing of the upstream ice flow relative to a homogenous case.

#### 1.2 Previous work

Previous research has provided much progress in understanding tidal flexure and its ice dynamic implications both in modeling and observations. In terms of observations, satellite coverage and resolution has proliferated in the last decades (e.g., Friedl et al., 2020) using altimeters (e.g., Li et al., 2022; Freer et al., 2023) and interferometric synthetic aperture radar imaging (InSAR, Rignot et al., 2019; Milillo et al., 2019; Milillo et al., 2022; Rignot et al., 2024). Most relevant for this study are differential InSAR applications (DInSAR, Rignot et al., 2011) which image tidal flexure even in regions with minimal tidal range such as the Ross Sea (Padman et al., 2018). As explained in more detail later, this technique captures the spatial extent of the tidal-flexure zone and reliably marks the landward and seaward limits of tidal flexure typically within fractions of the local ice thickness (Brunt et al., 2010). Modeling tidal flexure and its ice-dynamic implications is challenging, because all stress terms in the full Stokes equations might be relevant (Rosier and Gudmundsson, 2018), and also because tidal timescales require the consideration of elastic properties together with the viscous deformation (Reeh et al., 2000; Reeh et al., 2003; Gudmundsson, 2011; Wild et al., 2017). Nevertheless, this approach has led to important findings, for example in terms of the subglacial hydrology (Hewitt, 2013; Walker et al., 2013; Rosier and Gudmundsson, 2020) and stress coupling across the grounding line (Rosier and Gudmundsson, 2016).

The lateral shear margin of a glacier is a region of large deformation that separates the fast-flowing ice from stagnant ice or rock. Shearing within lateral margins leads to a resistive force (side drag) that counteracts a large proportion of the glacier driving stress. The nature of this side drag depends on local ice rheology, which changes as a function of damage,

temperature, melting and ice fabric, and these in turn depend on the local shearing (Minchew et al., 2018). Despite their important role in regulating glacier flow, shear margins are a relatively poorly understood component of glaciers due to the numerous challenges in accessing and measuring these complex regions.

#### 65 1.3 Current shortcomings

Rifts develop along lateral shear zones due to rapid shearing and localized strain (Alley et al., 2019). These surface rifts pose challenges for in-situ data collection (Arcone et al., 2016), limiting the availability of observational data needed to validate remote-sensing methods (Marsh et al., 2021). The situation is further complicated by significant surface velocity gradients, which cause phase decorrelation in InSAR processing. As a result, monitoring fast-flowing glaciers and ice streams, such as Thwaites Glacier (Milillo et al., 2019; Rignot et al., 2024) and the Dotson-Crosson Ice Shelf system (Milillo et al., 2022), is only possible via lower resolved SAR amplitude tracking when using publicly available satellite SAR data, which typically have repeat passes of multiple days to weeks. Addressing this issue requires dedicated satellite missions with shorter repeat acquisition intervals (hours to days) to ensure phase coherence for fast-flowing areas and particularly across shear zones. This limitation currently hinders an Antarctic-wide assessment. Future advancements in satellite radar technology are expected to enhance the temporal resolution of publicly available SAR data, improving their ability to capture the flexural response across shear zones and advancing our understanding of ice-shelf dynamics. It is therefore necessary to develop methods that infer spatially-variable ice properties indirectly.

# 1.4 Framework of this paper

Here we investigate if the heterogeneity in ice stiffness can be observed along the tidal-flexure zone, but instead of using the horizontal velocities as primary observations, we compare the observed vertical displacement with an elastic flexure model. We use a spatially variable Young's modulus as a proxy for both heterogeneous ice temperature, ice fabric and damage. Ice that is warmer and ice in which the ice fabric is aligned with the principal ice-flow direction will deform more strongly for a given stress regime than its cold and isotropic counterpart (Hudleston, 2015). In the elastic model we approximate this weakening with a lowered Young's modulus. As we will show later, a spatially heterogeneous Young's modulus leads to a non-local response in the prediction of tidal flexure, and misfits between predictions and observations can be used to diagnose such spatial heterogeneities from satellite observations.

The study is structured as follows: First, we test the hypothesis that reducing ice stiffness within lateral shear zones can exert a detectable signal in DInSAR mappings. If substantiated, this hypothesis implies that DInSAR imagery can offer insights beyond the conventional delineation of grounding lines, providing additional information about ice properties. Second, we use repeat-pass satellite measurements from DInSAR to constrain an elastic model of tidal flexure. Using the flexural response of floating ice near the grounding line to tidal forcing as a regional experimental setting, we reveal bulk softening in ice-shelf shear zones. This insight is derived from a comparison between homogeneous and heterogeneous model

experiments and satellite-derived tidal flexure patterns of Priestley Glacier. Third, we challenge our assumption of elasticity by incorporating in-situ data of surface deformation recorded both outside and within its tidal-flexure zone.

# 2 Hypothesis and study area

#### 2.1 Synthetic model experiments

We begin by conducting synthetic modeling experiments to illustrate the hypothesized effects of shear-margin weakening on the tidal-flexure patterns of an idealized outlet glacier. The synthetic ice thickness profile we use is broadly consistent with the known geometry of Priestley Glacier. Specifically, we assume ice thickness of 1000 m at the grounding line, tapering non-linearly to 500 m at the ice edge, following the model geometry described by Holland et al. (2008) and formulated by Walker et al. (2013):

$$H = \beta_1 / (\beta_2 - (x - x_{gl}))^{1/2}, \tag{1}$$

where H represents ice thickness, while parameters  $\beta_1$  and  $\beta_2$  are adjusted to control the ice thickness at the grounding line and ice edge. The model spans a domain of 12 km by 25 km, incorporating a sinusoidal grounding line to reflect the geometric features within a landward embayment (Fig. 1a).

We model the expected vertical deformation due to ocean tides with a peak amplitude of 1 m using an elastic model the details of which are explained in Sect. 3.3. Our baseline simulation to which all other simulations are compared assumes spatially homogeneous elastic properties represented by a constant Young's modulus (E). Spatial heterogeneity is introduced within the floating parts in the form of two localized regions with a lower Young's modulus on either side of the model domain (Fig. 1b). Contour lines of vertical displacement for any time interval in a tidal cycle are visualized with a cyclic colormap (Figs. 1 c and d; here specifically one color cycle, or one fringe, corresponds to 10 cm). Such colormaps are commonly used in satellite interferometry which we will use later as our main source of observations.

We find that introducing elastic heterogeneity leads to a non-local response in tidal flexure which is expressed in difference to the homogeneous case in the form of a distributed bulge most prominently in the ice-shelf center (Fig. 1e). More specifically, the degree of hydrostatic balance (i.e., the ratio of ice deformation over tidal deformation denoted as ' $\alpha$ ' where 0 % depicts fully grounded and 100 % depicts fully floating ice) changes with the strength of heterogeneity (Fig. 1f). This suggests that the tidal flexure pattern can be used as a proxy to extract bulk heterogeneity. The magnitudes involved (typically 

Figure 1: Hypothesis: (a) Idealized ice thickness for synthetic model experiments. (b) The Young's modulus field, where values are varied in the shear-zone areas (shown in blue) in steps of 10 % from 10 % to 100 % of the Young's modulus used for the remaining ice-covered areas (indicated in yellow), which is set at 1.6 GPa. (c) Modeled tidal flexure under 1m tidal forcing for a homogeneous case, and (d) a heterogeneous case. (e) Their difference, showing a bulge up to 10 % of the applied tidal forcing. (f) Cross-section along the centerline (dashed lines in panels c and d), illustrating the flexure curves where Young's modulus is reduced in lateral shear zones. Panels (g) and (h) depict vertical tidal flexure during high and low tide scenarios. Each color fringe represents approximately 10 cm of vertical displacement, corresponding to a tidal amplitude of ±60 cm in this schematic. The concentrated band of tidal fringes marks the grounding line at its landward edge and the hydrostatic line at its seaward boundary. Beyond this, the ice floats freely on the ocean, oscillating harmonically with the tides. Note that vertical displacement is not to scale in this schematic.

# 2.2 Study area

Priestley Glacier is a 96 km (60 mi) long outlet glacier in Victoria Land, East Antarctica, originating from the edge of the polar plateau. It flows through the Transantarctic Mountains, draining into the fjord-like grounding zone of the Nansen Ice Shelf in Terra Nova Bay (Fig. 2). The glacier features floating shear zones with long-term shear strain rates around 0.01 yr<sup>-1</sup> (Fig. S1). A stake network survey conducted on the glacier's true left shear zone revealed spatial variability in ice stiffness, with softer ice in the shear zone and stiffer ice in the central part of the glacier (Still et al., 2022). Observations also identified localized tidal flexure, which could only be modeled by incorporating a spatially heterogeneous Young's modulus or modifying the boundary conditions to account for coupling with a rocky ridge on the glacier's left sidewall. Ice core measurements taken from the same area showed that grain size reduction and preferential c-axis alignment, consistent with shear deformation along a vertical shear plane perpendicular to the flow direction, contributed to the mechanical ice softening (Thomas et al., 2021).

Figure 2: Landmarks along the Victoria Land coast of East Antarctica. (a) The dashed-red rectangle shows the location of the study area in panel b. (b) The transitioning zone of Priestley Glacier, draining into the Nansen Ice Shelf, superimposed with a velocity field derived from TerraSAR-X data in 2018. The map background is the Landsat image mosaic of Antarctica (LIMA, Bindschadler et al., 2008). The grounding line is sourced from the Antarctic Surface Accumulation and Ice Discharge project (ASAID, Bindschadler et al., 2011). The inset shows the location of the study site along the Transantarctic Mountains in East Antarctica. Map coordinates in an Antarctic Polar Stereographic projection (EPSG:3031).

# 3 Data and methods

We seek to detect differences between modeled and observed vertical displacements throughout the tidal cycle. This requires adjustments of the tidal forcing used for predictions to overcome tide-model inaccuracies, and a decomposition of the double-differential tidal displacement measured by DInSAR into a synthesized time series of 'single-tide' displacements at the times of SAR data acquisitions. The process begins with deriving an  $\alpha$ -map from DInSAR measurements, representing the average tide-deflection ratio. Next, the tide model output is adjusted to align with DInSAR observations over the freely-floating portion of the ice shelf. This adjusted tide model is then combined with the alpha map to predict average tidal displacements at the times of SAR data acquisition.

160

These predicted tidal displacement maps are reassembled to compute residual mismatches with the original DInSAR images. Similar to the tide model adjustment, we then determine the minimal per-pixel offsets needed in the average tidal displacement maps to best match the DInSAR data. By subtracting these offsets from the predicted tidal displacement maps, we effectively decompose the DInSAR images into their underlying 'single tide' components, enabling the reconstruction of synthetic displacement maps corresponding to each SAR data acquisition.

When these synthetic maps are again reassembled for comparison with the original DInSAR images, we refer to the resulting products as Mosaics. In the following we detail the individual steps for Priestley Glacier, an example for Darwin Glacier can be found in Wild et al., 2019 To assess the role of shear-zone weakening, we replace the  $\alpha$ -map from DInSAR measurements with modeled  $\alpha$ -maps from homogeneous and heterogeneous experiments. We validate model solutions with GPS data from two sites and find that incorporating any  $\alpha$ -map significantly improves the fit between observed and modeled displacements. However, the GPS sites themselves are not particularly sensitive to the specific modeling choices, which motivates our broader evaluation across the entire tidal-flexure zone. Finally, the GPS records are also used to estimate the timing of tidal oscillations at the two sites to assess the validity of the elastic approximation.

## 180 3.1 Remote-sensing of the tidal-flexure zone

# 3.1.1 Satellite-based surface velocity and DInSAR tidal flexure

TerraSAR-X is the highest resolution sensor (

$$D = \frac{EH^3}{12(1-\lambda^2)},$$
 (4)

$$250 \quad q = \rho_{sw} g(A - w), \tag{5}$$

where E(x,y) is the effective Young's modulus for ice. We use the term "effective" Young's modulus to acknowledge that, in natural, real-world conditions, ice behaves differently from theoretical or laboratory-based conditions due to factors like ice fabric, temperature variations, and impurities. H(x,y) represents ice thickness,  $\lambda$ =0.4 is Poisson's ratio accounting for lateral deformation due to longitudinal strain,  $\rho_{sw}$ =1027 kg m³ is ocean water density, g=9.81 m s³ is the gravitational acceleration, and A(t) is the time-varying, DInSAR adjusted, tidal forcing. The boundaries of the grounded portion in the model domain were rigidly anchored (w=0,  $\nabla^2 w$ =0), while the ice-shelf edge moves synchronous with tides (w=A(t),  $\nabla w$ =0). To facilitate stress transmission between the floating and grounded portions, a fulcrum was used at the grounding line (w=0). Similar to previous studies (Wild et al., 2017, 2018, 2019), the equations were solved with a finite-element solver implemented in COMSOL Multiphysics using a triangular mesh. The equations were numerically integrated with a fully implicit time-stepping scheme (backward differentiation formula).

Unless otherwise stated, we use E=1.6 GPa for our reference scenario which is motivated by tiltmeter measurements across an unconstrained section of the grounding line of the Southern McMurdo Ice Shelf (Wild et al., 2017, 2018). The value of the effective Young's modulus dominates the offshore flexure pattern, while the bed stiffness controls the amount of grounded ice flexure (Appendix A.2) We neglect viscous deformation on tidal timescales (Reeh et al., 2000, 2003), an assumption that we discuss in Sect. 5.2. This model setup was also used in a synthetic geometry to motivate our hypothesis (Sect. 2.1), next we detail the real world scenarios at Priestley Glacier.

# 3.3.2. Input variables and forcing for modeling tidal deformation at Priestley Glacier

Common to all simulations is the tidal forcing which is based on the Circum-Antarctic Tidal Simulation model (CATS2008, Howard et al., 2019, an updated version of the model by Padman et al., 2002) to predict tidal oscillations at a location on the freely-floating area of Priestley Glacier. To correct for the weight of the tidal wave on the Earth's crust, we assumed an elastically deforming bed and used the global barotropic assimilation model from Oregon State University to calculate load tides (TPXO9, Egbert and Erofeeva, 2002). Additionally, we corrected for the inverse barometric effect (IBE, Padman et al., 2003) using ERA-5 reanalysis data (Bell et al., 2020). Thereby, a +1 hPa anomaly of atmospheric pressure translates to an instantaneous -1 cm contribution to the tidal forcing. We validated pressure variations with an available 59-day record of barometric pressure and found excellent agreement (Pearson correlation: 0.96). In this paper, "tidal forcing" refers to the combined outputs of ocean tide model, load tide model, and the IBE. The tidal forcing was applied only to floating ice, with signals such as mean dynamic topography and storm surges, which cannot be directly measured at the ice-shelf surface, neglected.

#### 280 3.3.3. Model experiments

# 3.3.3.1 Control model setup

The Control model represents the model setup based on publicly available input data sets for the Priestley Glacier. We used the ASAID grounding line (Bindschadler et al., 2011) derived from Landsat-7 data (1999–2003) and ICESat data (2003–2008), and the BedMachine ice thickness (Morlighem, 2020; Morlighem et al., 2020). In the Control model setup (Fig. 3a) the Young's modulus is homogeneous at 1.6 GPa.

#### 3.3.3.1 Local model setup

285

290

In the Local model setup we incorporated an updated ice thickness map from more recent inversion of ice-shelf freeboard and a grounding line delineated from one TerraSAR-X DInSAR image (Fig. 3b). For the thickness update, we used freeboard inversion with surface elevation from the Reference Elevation Model of Antarctica (REMA, Fig. S3a, Howat et al., 2019) with the EIGEN6c4 geoid model (Fig. S3b, Foerste et al., 2014) as the mean sea level. For grounded areas, we used the BedMachine ice-thickness product (Morlighem, 2020; Morlighem et al., 2020). No firn correction is needed for the blue ice surface of Priestley Glacier. The biggest differences of the local model and the control model emerge from the inferred ice thickness at the grounding line, with differences of up to 150 m (Fig. S4).

Figure 3: Setups for the Control and Local model at Priestley Glacier. (a) The Control model uses the ASAID grounding line and BedMachine v2 ice thickness. (b) The Local model uses a DInSAR-derived grounding line from TerraSAR-X data and ice thickness from freeboard inversion for floating ice. Colored outlines indicate boundary conditions. The finite-element mesh is in the background. The dashed gray line shows the IceBridge flight path (Nov 2013) with radar data for ice-thickness validation (Appendix A). Dashed blue and orange lines represent locations of homogeneous model solutions in Figure 9 j and k, respectively.

#### 4) Results

305

# 4.1 Tide-model accuracy before and after adjustment

At Priestley Glacier, modeled ocean tides exhibit the highest amplitude variability, reaching  $\pm 0.4$  m, followed by the IBE at  $\pm 0.15$  m, and the load tide at  $\pm 0.05$  m (Fig. 4a). The combined effect, referred to as tidal forcing, reaches  $\pm 0.6$  m. The tides exhibit mainly diurnal characteristics with a 14-day spring-neap cycle. The correlation between ERA5 reanalysis data and a 59-day pressure record from a local AWS at Priestley Glacier is 0.96 (Fig. 4b), indicating high confidence in the IBE correction. We use these time series to predict tidal forcing for the 56 Sentinel-1 SAR data collection times (Fig. 4c and d). The mapped vertical displacements of the 56 SAR acquisitions cover a time interval of more than a year and all stages of the tidal oscillation.

Figure 4: Tidal forcing at Priestley Glacier. (a) Tidal forcing combines the CATS2008 tide model, TPXO9v2a load model, and the IBE from ERA5 data, compared to an IBE derived from a local AWS. Shaded areas show periods with in-situ data for validation. (b) Detailed view of November–December 2018, when AWS and GPS data were collected. (c) Predicted tidal forcing at Sentinel-1 SAR acquisition times. (d) Focused view of November–December 2018. Gray lines indicate SAR acquisition dates; red dots show tidal forcing values from the model.

Prior to the adjustment with DInSAR observations in the freely floating areas (Fig. 5a), the misfits between the modeled and measured differential tidal displacement is  $\pm 0.15$  m (Fig. 5b). This is larger than the general accuracy of tide models in Antarctica ( $\pm 6.5$  cm, Stammer et al., 2014). After the adjustment of the modeled tidal forcing (Methods, Section 3.1.3.), the misfit is reduced to effectively zero, resulting in a match with all measurements (Fig. 5c). The required offsets at the 56 time stamps to achieve this greater accuracy have a mean of 0.04 m and a temporal standard deviation of 0.03 m.

Figure 5: Sentinel-1 DInSAR measurements and tide model comparison at Priestley Glacier. (a) Vertical displacements along an IceBridge profile in the tidal flexure zone from 31 DInSAR interferograms. Colored dots show tidal forcing on the ice shelf; dashed gray lines mark the Tuati (flexure zone) and Shirase (floating) GPS stations. (b) Measured vs. modeled tidal forcing (colored dots), with the black dashed line indicating a perfect match. (c) Comparison after adjusting the modeled tidal forcing to align with DInSAR measurements.

The mapping of vertical displacement to individual SAR scenes shows smooth vertical flexure for most stages of the tidal oscillation (Fig. 6). Image quality is overall high, with the exception of a narrow band close to the lateral shear zones which show low coherences throughout the dataset.

Figure 6. Selection of twelve synthesized vertical displacement maps in the tidal flexure zone of Priestley Glacier, showing vertical displacement due to ocean tides during SAR data acquisitions in 2017. The dashed line indicates the IceBridge flight path (Nov 2013). Red and orange dots mark the GPS stations Shirase (freely-floating) and Tuati (within the tidal flexure zone).

With the 56 synthesized displacement maps corresponding to the SAR data acquisition times, we can now generate any desired combination of images. We reassembled the 31 double differences corresponding to the DInSAR combinations to enable a more direct comparison between the measured and mosaicked interferograms. The synthetic interferograms replicate the complex tidal fringes observed in the DInSAR measurements, with discontinuities appearing only outside of the central region of Priestley Glacier (Fig. 7).

Figure 7: Selection of three measured and mosaicked images from 31 available DInSAR combinations showing double-differential vertical displacement. The SAR image combination is indicated in the lower left corner. The dashed line marks the IceBridge

flight path from November 2013, with red and orange dots showing the locations of the Shirase (freely-floating) and Tuati (tidal flexure zone) GPS stations.

#### 345 4.2 Effect of ice heterogeneity in lateral shear zones

350

355

In the process of DInSAR image mosaicking, we initially used an empirical  $\alpha$ -map derived only from all Sentinel-1 DInSAR images. However, alternative  $\alpha$ -maps, generated from elastic finite-element model solutions, can also be used to create synthesized vertical displacement maps. This strategy allow us to evaluate the effect of reducing the effective ice stiffness in the shear zone on the tidal flexure pattern by using modeled  $\alpha$ -maps where the Young's modulus was locally reduced. The goal is to achieve the best possible match between the mosaicked maps and the root-mean-square error (RMSE) between the measured and mosaicked DInSAR images (Fig. 8a).

We tested three distinct finite-element model configurations: (i) Control model, using input data from publicly available datasets, (ii) Local homogeneous model, incorporating ice thickness from freeboard inversion and DInSAR grounding line but assuming uniform ice properties across the modeling domain, and (iii) the Local heterogeneous model, sharing the same input as (ii) but featuring ice stiffness variability where the Young's modulus is systematically reduced in lateral shear zones. We use the Control model as a test for its applicability for a continent-wide assessment, where updated grounding lines and ice thicknesses are not yet available.

All model configurations exhibit minimal RMSE in the central region of the floating ice shelf (Fig. 8b-d), indicating that they align well with the mosaic (Fig. 8e-g), irrespective of the α-map used. This demonstrates that our least-squares adjustment routine, as detailed in Sect. 3.1.4, is effective regardless of the applied α-map. However, variations within the shear zone are random (Fig. 8e-g) and stem from phase decorrelation in these areas. Therefore, to gain a clearer understanding of the impact of ice stiffness on the tidal flexure pattern, it is more insightful to directly compare the measured and modeled α-maps rather than focusing solely on the RMSE between the measured and modeled DInSAR images.

Figure 8: RMSE between 31 DInSAR images measured with Sentinel-1 and least-square predictions based on various  $\alpha$ -maps. Panel (a) presents the mosaic using an empirically-derived  $\alpha$ -map from Sentinel-1 data. Panels (b) to (d) show results from different finite-element model setups. Panels (e) to (g) compare the mosaicked and modeled RMSE, showing that all model setups replicate the mosaic accurately within noise levels (

Figure 9: Comparison of percentage tidal displacement determined by various  $\alpha$ -maps. Panels (a) to (d) show tidal displacement percentages: (a) measured from 31 Sentinel-1 interferograms, (b) from the Control model, (c) from a homogeneous model, and (d) from a heterogeneous model with 20 % effective stiffness in the lateral shear zones. Black contours outline the hydrostatic line, where the ice is in hydrostatic equilibrium, exhibiting full tidal motion ( $\alpha$ =100%). The dashed gray line marks the IceBridge flight path, indicating flexure curve locations in panels (i) to (l). Panel (e) shows measured tidal fringes in Priestley Glacier's grounding

zone, highlighting phase decorrelation in the shear zones. Panels (f) to (h) present differences between measured and modeled α-maps, with red and orange dots marking GPS receiver locations on the floating and tidal flexure zones. The green hatched region marks the region analysed in the scatterplots in Figure 11. Panels (i) to (l) show flexure curves, with the heterogeneous model (20 % E in lateral shear zones) closely matching the measured pattern. Dashed lines mark the locations of the Tuati (orange) and Shirase (red) GPS stations.

#### 390 4.3 Validation with GPS records

We validate our findings using in-situ measurements from two GPS sites: Shirase, which is nearly freely floating, and Tuati, located in the tidal flexure zone. At Shirase, the mean ice-flow speed is 105 m yr<sup>-1</sup>, while at Tuati, it is 131 m yr<sup>-1</sup>. These speeds are consistent with TerraSAR-X-derived measurements of 96 m yr<sup>-1</sup> and 132 m yr<sup>-1</sup>, respectively. To isolate the vertical tidal signal, we remove linear trends of -0.81 m yr<sup>-1</sup> at Shirase and -2.44 m yr<sup>-1</sup> at Tuati from the GPS elevation records. We then compare the resulting anomalies with tidal forcing predicted using various  $\alpha$ -values from different models (Fig. 10).

The measured  $\alpha$ -value for the Shirase GPS site is 89.6 %, which is overestimated by all model setups (Control model: 105.8 %, Local homogeneous: 101.5 %, Local heterogeneous: 104.0 %). For Tuati, the measured  $\alpha$ -value is 17.5 %, with the best match provided by the Local heterogeneous model (Control model: 9.8 %, Local homogeneous: 13.2 %, Local heterogeneous: 16.2 %).

Using these  $\alpha$ -values, we linearly scale the previously derived tidal forcing to predict vertical tidal motion at the GPS locations. Including tidal loading and the IBE reduces the RMSE between the unscaled tidal forcing and the freely-floating Shirase GPS measurements from 0.076 m to 0.067 m. Scaling with the measured  $\alpha$ -value improves the RMSE to 0.064 m. However, the modeled  $\alpha$ -values slightly worsen the fit for Shirase (Control model: 0.071 m, Local homogeneous: 0.068 m, Local heterogeneous: 0.07 m).

The benefits of using an  $\alpha$ -map are more pronounced in the spatially extended tidal flexure zone. The RMSE between the unscaled tidal forcing and the Tuati record is 0.182 m. Scaling with the measured  $\alpha$ -value of 17.5 % improves the fit to 0.03 m (or by 84 % ). The various modeled  $\alpha$ -maps offer slightly better fits (Control model: 0.027 m, Local homogeneous: 0.027 m, Local heterogeneous: 0.029 m). Therefore, as expected, any  $\alpha$ -value scaling improves the fit between observed and predicted tidal displacements. The differences between using the  $\alpha$ -values from our three model scenarios are small and show only comparatively little variability across the different model scenarios. This means that the locations of Shirase and Tuati are not particularly sensitive to the modelling choices, which is also evident in Figs. 9f-h. However, differences between the different model scenarios are more pronounced when evaluated across the entire flexure zone as opposed to along the central flowline only.

Figure 10: Example of improved tide-model output over one neap-spring-neap tidal cycle, achieved through linear scaling with the tide-deflection ratio from the Control model. (a) The components of the tidal forcing. (b) Tide modeling accurately predicts measured vertical displacement where ice is freely floating, but (c) neglects the damping effect of ice dynamics in the tidal-flexure zone if unscaled. The locations of the two GPS receivers are shown in Figure 9.

In the tidal flexure zone (green hatched region in Fig. 9f-h), Local heterogeneous models with a Young's modulus ranging from 30% to 100% consistently underestimate the satellite-derived percentage of tidal displacement. The optimal fit is obtained with a 20% Young's modulus. A 10% Young's modulus tends to overestimate the tidal flexure, particularly for high  $\alpha$ -values near the hydrostatic line (Fig. 11). The RMSE for the homogeneous case is 14.8%, and for the best heterogeneous case is 10%, representing an improvement of 33%. Further reducing the Young's modulus from 20% of its reference value to 10% increases the mismatch again, raising the RMSE to 10.4 %.

Figure 11: Scatterplots comparing measured and modeled  $\alpha$ -maps from the Local heterogeneous model of the tidal flexure zone near Priestley Glacier's grounding line. Point density is color-coded, with yellow denoting high density. The red dashed line represents a perfect match. The heterogeneous model, using a 20 % Young's modulus in the shear zone, aligns best with measured values. The RMSE for each scatterplot is minimal for 20% E, and maximal for the homogeneous case (100 % E).

# 5) Discussion

# 5.1 Hypothesis evaluation: Ice stiffness reduction in shear zones

Our results indicate that heterogeneous flexure models provide a better fit to DInSAR observations than their homogeneous counterparts. This supports the idea that satellite interferometry is sensitive to spatial variations in ice stiffness, particularly across shear zones. However, this interpretation depends on the choice of the reference Young's modulus, which we set to E = 1.6 GPa. To explore the sensitivity of our findings to this assumption, we conduct additional simulations using the homogeneous model. We find that lowering the homogeneous Young's modulus to E = 1.0 GPa yields a fit to the DInSAR measurements comparable to that of the best-fit heterogeneous model, which assumes E = 1.6 GPa with a 20% reduction across shear zones (Fig. 12). This non-uniqueness highlights the importance of better constraining field-derived estimates of the effective Young's modulus.

Figure 12: Comparison between the best homogeneous and best heterogeneous flexure model solutions. (a) Elastic flexure for a homogeneous Young's modulus of E = 1.0 Gpa. (b) Difference relative to the homogeneous reference case of E = 1.6 Gpa. (c) Flexure curves for both homogeneous cases compared to the DInSAR measurements and the Control model. (d) Elastic flexure for a heterogeneous ice stiffness model with E = 1.6 GPa and a 20% reduction in the lateral shear zones. (e) Difference between the best heterogeneous case and the homogeneous reference (E = 1.6 Gpa). (f) Best-fit heterogeneous flexure curve compared to satellite observations and the Control model.

The reference value of E = 1.6 GPa is based on previous tiltmeter measurements from a relatively simple setting on the Southern McMurdo Ice Shelf, which features a thick firn layer and limited lateral variability (Wild et al., 2017). In contrast, the Priestley Glacier contains well-defined shear zones, consists largely of blue ice, and originates from the polar plateau, where ice temperatures are significantly colder. Given these differences, we expect the Priestley Glacier ice to be inherently

stiffer than that of the Southern McMurdo Ice Shelf. For this reason, we retain E = 1.6 GPa as the reference value for our heterogeneous experiments and do not adopt the reduced homogeneous value as a new baseline.

This result is reminiscent of findings from the Darwin Glacier, where tiltmeter data indicated a Young's modulus of E = 1.0 GPa (Wild et al., 2019). Like Priestley Glacier, Darwin Glacier exhibits a pronounced lateral shear zone and is composed predominantly of blue ice. These similarities raise questions about the appropriateness of using a reduced Young's modulus in such settings, where colder ice temperatures and structural complexity should lead to higher stiffness.

#### 5.2 Elastic weakening in shear zones: Contradictions and interpretations

In Wild et al., 2019, we identified a narrow band of relatively larger model misfits stretching from Darwin Glacier's shear margin to the freely-floating ice shelf. These discrepancies were attributed to the microscopic process of ice anisotropy, to explain the observed macroscopic response and accounted for the glacial heterogeneity within the embayment. In the present study, we attempted to replicate a similar signal within the shear zone of Priestley Glacier using Sentinel-1 data. However, phase-decorrelation in the shear zone impeded a direct enhancement of the model fit in these lateral regions.

However, the core issue is that we observe apparent elastic weakening within shear zones, which stands in contrast to laboratory measurements indicating that ice anisotropy has only a minimal effect on the elastic properties of glacier ice (Lutz et al., 2022; Rathmann et al., 2022). This suggests that attributing reductions in Young's modulus within shear zones to ice fabric is not physically justified. Our efforts also rely on isotropic material assumptions, which do not align with natural ice masses (Jacka and Budd, 1989). In regions characterized by simple shear, such as the lateral shear zones of fast-flowing outlet glaciers, there is a notable transition from initially isotropic to anisotropic ice (Thomas et al. 2021). This transformation results in a robust fabric aligned perpendicular to the shear plane, primarily due to the basal glide deformation of individual crystals. As grain rotation and dynamic recrystallization take place within the polycrystal, a preferred orientation fabric emerges, leading to bulk softening of ice stiffness. In polycrystalline shear zone ice with well-developed fabrics, the enhancement of ice flow can be up to ten times greater than in the isotropic case (Hudleston, 2015). This becomes significant in regions where fast-flowing glaciers are constrained along their margins, leading to the disappearance of basal drag, and shear stresses from lateral boundaries become the sole determinant of effective strength to resist grounding-line discharge (Gudmundsson, 2013).

More broadly, the contradictions between field- and laboratory derived properties call into question the appropriateness of using Young's modulus as a bulk tuning parameter to represent the combined effects of fabric, temperature, and damage. While ice fabric has limited influence on elastic properties, it significantly affects the viscous behavior of glacier ice. Given that tidal flexure is fundamentally a viscoelastic process, it would be more physically consistent to vary ice viscosity, rather than elasticity, within shear zones when tuning models to match observations. Nevertheless, because our analysis is based on

comparisons of modeled and observed  $\alpha$ -maps, which are temporally averaged and thus largely filter out transient viscoelastic signals, we sidestep the need to explicitly implement a viscoelastic model in this study.

This  $\alpha$ -map based approach is only feasible because a long time series of Sentinel-1 data is available, capturing the full tidal cycle and enabling a reliable temporal average of the flexural response. As publicly accessible SAR satellite data becomes more abundant, we anticipate significant progress in our understanding of shear-zone weakening. This advancement will enhance the applicability of the methods outlined in this study, potentially informing the next generation of ice-sheet models with improved maps of spatially heterogeneous ice properties along Antarctica's grounding line.

## 5.3 Limitations and opportunities for improving shear zone characterization

Caveats of our method is the reliance on synthesizing 'single-tide' displacement maps, and limited interferometric coherence to resolve fringes across the shear zone. These, in turn, hamper our capacity to unequivocally distinguish between potential mechanisms of shear zone weakening—such as anisotropy, shear heating (Perol and Rice, 2015), fracturing (Albrecht and Levermann, 2014), or nonlinear viscous effects governed by Glen's flow law under tidal loading (Rosier and Gudmundsson, 2018). To definitively link the observed weakening in the shear zone of Priestley Glacier to the macroscopic influence of ice-crystal anisotropy, more in-situ data are required. These data should not only capture 'single-tide' vertical flexure, but also resolve fringes across the shear zone, particularly where Sentinel-1 data lose sufficient coherence. It remains unclear whether the phase-decorrelation observed in the shear zone of Priestley Glacier stems from intrinsic differences in setting compared to Darwin Glacier, or if it is related to the use of Sentinel-1 data, which differs from the TerraSAR-X data used in our previous study. Despite this limitation, we successfully matched tidal flexure in the central areas of Priestley Glacier, providing evidence for shear-zone weakening and its macroscopic influence on the flexural pattern.

Terrestrial radar interferometry, as applied by Drews et al. (2021) to the upstream section of Priestley Glacier, could fill this data gap due to its significantly higher temporal resolution (minutes to hours) compared to satellite SAR data (days to weeks). With tidal fringes extending across the shear zone, it would be possible to better constrain viscoelastic models in key regions affecting Priestley Glacier's resistance to vertical displacement, than using its central part as a proxy for processes occurring along its sides.

# 5.4 Justification for using an elastic approximation

The RMSE provides a useful measure of the absolute fit between the modeled tidal forcing and GPS records. However, it does not offer information about the timing of tidal oscillations at the two sites. To address this, we examine the correlation between unscaled tidal forcing and the GPS records. For Shirase, including load tides and the IBE improves the correlation from 0.92 to 0.94. For Tuati, the correlation increases from 0.62 to 0.66. The difference in correlation between the two GPS sites, about 0.3, suggests varying timing of tidal oscillations, which we attribute to viscoelastic effects. In an ideal elastic

scenario, tidal deformation would occur instantaneously, leading to consistent correlations across the tidal flexure zone, though with a reduced tidal amplitude through flexural damping.

Figure 13 illustrates the effect of introducing a synthetic time shift on the correlation between tidal forcing and GPS measurements. Applying a 0.37-hour shift to the unscaled tidal forcing improves the correlation with the Shirase GPS from 0.94 to 0.95. This minor improvement supports the assumption of hydrostatic equilibrium at the freely-floating GPS site. In contrast, applying a 1.3 h time shift enhances the correlation between the flexure-zone GPS and tide model output from 0.66 to 0.72, suggesting that viscoelastic time delays occur between the two GPS sites. Although estimating ice viscosity for a more precise fit is theoretically feasible, the elastic model used in this study, achieves an 84 % improvement compared to a tide model excluding tidal flexure, and provides a substantial enhancement. Further exploration of viscoelastic modeling is planned for future research.

Figure 13: Correlations and associated RMSE between GPS measurements and tide model output as a function of a synthetic time shift. The length of the orange arrow corresponds to a 1.3 h time delay that we largely attribute to viscoelastic time delays in the flexural response between the two GPS sites.

To approximate tidal flexure with an elastic model, significant tidal amplitudes during SAR image acquisition are crucial to minimize viscoelastic time delays (Wild et al., 2017). Instead of selecting individual DInSAR images for large tidal amplitudes, we reduce viscoelastic effects by computing a percentage tidal displacement field over all 31 Sentinel-1 DInSAR images ( $\alpha$ -map; Han and Lee, 2014; Wild et al., 2019). This averaging approach also addresses geometric effects from horizontal surface motion around a neutrally-deforming layer at the center of the ice shelf (Suppl. Video, Rack et al., 2017; Wild et al., 2018). Generating an  $\alpha$ -map assumes minimal tidal grounding-line migration, as observed at Priestley Glacier, where high-temporal resolution terrestrial radar interferometry indicates horizontal grounding-line migration well below the scale of one ice thickness (Drews et al., 2021).

#### 5.5 Asymmetric tides and potential role of ephemeral re-grounding

Our GPS data show an asymmetric ocean tide, with a noticeably steeper rising tide compared to the falling tide (Fig. 10b). This asymmetry is not captured by the tide model, which predicts a more balanced, symmetric tidal oscillation. There are a couple of potential explanations for this discrepancy: (1) preferential propagation of the tidal wave, which might enhance tidal inflow into the sub-ice-shelf cavity more than the outflow during the falling tide, or (2) transient re-grounding of the ice shelf at low tide. The latter has been suggested by Drews et al. (2021) and could be linked to upstream flexure opposing the direction of the falling tide. This opposing flexure may suggest the formation of a temporary fulcrum offshore of the grounding line, potentially aligning with undulations in the bathymetry. However, our tidal flexure model currently does not account for the shape of the offshore bathymetry, preventing us from evaluating the effects of transient re-grounding. Although it would be theoretically possible to incorporate the bathymetry as an elastically deforming boundary condition—similar to the bed beneath grounded ice in the current model setup—the limited availability of bathymetric data in the tidal flexure zone complicates this modeling effort (Appendix A).

# 6) Conclusions

This study highlights the importance of accounting for ice heterogeneity and tidal flexure in modeling the dynamics of Antarctic outlet glaciers. By integrating Sentinel-1 DInSAR data with tidal-flexure models, we demonstrated that reducing ice stiffness in lateral shear zones significantly improves the accuracy of vertical displacement predictions, particularly along the grounding line. Questions remain regarding the choice of the reference Young's modulus. In our heterogeneous experiments, we use E = 1.6 GPa, while in the homogeneous experiments we adopt E = 1.0 GPa, providing an alternative and equally valid explanation for the observed behavior. This highlights the need for better constraints on field-derived estimates of the effective Young's modulus. Nonetheless, we emphasize the broader utility of DInSAR beyond its conventional role in delineating grounding lines of ice shelves to map spatially variable ice properties—a first step toward mapping ice rheology along the Antarctic coastline to inform ice-sheet models.

Incorporating DInSAR into large-scale ice-sheet monitoring not only enhances our understanding of glacial dynamics but also provides a critical tool for mapping the mechanical properties of ice, essential for future studies on ice rheology and deformation. Expanding this approach to other Antarctic outlet glaciers, combined with the increasing availability of satellite data, will further refine our knowledge of ice heterogeneity and its impact on glacier flow, potentially transforming how we predict ice-sheet responses to a changing climate.

# 575 Appendix A

#### A.1 Ice-thickness validation with airborne radar

Figure A1 illustrates a radar transect along an IceBridge flight path acquired in November 2013 (Paden et al., 2010), covering Priestley Glacier as it merges with the Nansen Ice Shelf. The REMA surface elevation data closely matches the airborne measurements from IceBridge. However, discrepancies are more pronounced in the ice base measurements. Upstream of the grounding line, marked by a distinct reduction in basal reflectivity on the IceBridge radargram (black line in Fig. A1), the freeboard inversion method significantly overestimates ice thickness by several hundred meters. Consequently, the BedMachine ice thickness product is employed for both the Control and Local models in this region.

In contrast, downstream of the grounding line, the freeboard inversion ice thickness estimates align well with the airborne measurements. Here, the ice thickness decreases from approximately 1000 meters at the grounding line to about 250 meters near the ice-shelf front, consistent with the hydrostatic equilibrium assumed in this portion of the Nansen Ice Shelf.

Figure A1: IceBridge radar profile. The cross-section shown in (a) follows the IceBridge flight path and presents (b) the airborne measurements of ice surface and ice base elevations (blue lines). REMA data indicates the ice surface elevation, while the inversion of freeboard is used to determine the ice base depth (dashed red lines). Mean sea level is referenced against the EIGEN-6c4 geoid (dashed green). The approximate location of the grounding line is marked in black.

#### A.2 Sensitivity tests

Our analysis indicates that the Young's modulus in the shear zones is reduced by a factor of five. To assess whether this effect stems from our model assumptions or represents a robust feature of the data, we perform a number of model experiments. Specifically, we investigate how varying the homogeneous Young's modulus and decreasing the foundation's spring stiffness beneath the grounded ice influence the modeled elastic response in our idealized setup. This setup assumes an idealized ice thickness and a grounding-line embayment resembling the geometry of Priestley Glacier, as described in Section 2.1.

Figure A2: Sensitivity of tidal flexure to varying Young's modulus and bed stiffness. (a/c) The range of tidal flexure for Young's modulus/bed stiffness from 20% to 100% as a function of grounding-line geometry. (b/d) Tidal flexure curves for both increasing and decreasing the reference values. (e) Zoom into the upstream bulge area, with the inset showing vertical displacement in centimeters along the y-axis

We systematically vary the homogeneous Young's modulus around its reference value of 100 % E = 1.6 Gpa and find that reducing the Young's modulus, i.e. softening of the ice, steepens the modeled tidal-flexure curve for a given tidal amplitude of A = -1m. This steepening is most pronounced at the floating sections of a grounding-line embayment (Fig. A2a), and moves the offshore inflection point, where tidal flexure over-shoots the applied forcing closer to the grounding line (Fig. A2b).

Reducing the foundation stiffness to 5% of the reference value (k = 5 MPa m<sup>-1</sup>) increases the flexure on both the floating and grounded portions of the ice. The differences between 100% k and 20% k are most pronounced at the grounded sections of adjacent grounding-line protrusions and least pronounced along the glacier centerline (Fig. A2c). Along the centerline, the modeled flexure curve shows a mean range of  $0.3 \pm 0.3$  cm on the grounded portion and  $1.6 \pm 1.1$  cm on the floating portion, measured from the grounding line to 10 km in the along-flow direction (Fig. A2d). This corresponds to a maximum of 2–3% of the applied 1 m tidal forcing, which is significantly smaller than the 10% bulge observed between the heterogeneous and homogeneous cases.

Interestingly, the inland extent of the grounding-line bump (Fig. A2e) is also influenced by grounding-line geometry, with protrusions exhibiting a greater inland reach than the glacier centerline (Fig. A2c). This occurs because grounding-line protrusions have more surrounding floating ice available to induce flexure on the grounded part compared to a grounding-line embayment.

#### 620 Competing interests

RD is a member of the editorial board of The Cryosphere.

#### Acknowledgements

CTW was supported by the Deutsche Forschungsgemeinschaft (DFG) in the framework of the priority programme 1158 'Antarctic Research with comparative investigations in Arctic ice areas' by a grant (DR 822/8-1). CTW, OJM, and WR

gratefully acknowledge the support from a NZ Antarctic Research Institute Type-A Grant 2018-1. RD acknowledges support from an Emmy Noether Grant (DR 822/3-1). NN is grateful for the support received from the BMBF IceSense (03F0866A) project. WSL was supported by the Korea Institute of Marine Science & Technology Promotion(KIMST) funded by the Ministry of Oceans and Fisheries(RS-2023-00256677). Special thanks to Antarctica New Zealand and the Korea Polar Research Institute for providing logistical support during the event K050 from Jang Bogo Station in the 2018/19 season. We express our appreciation to Patrick Power, mountaineer Richard Bottomley, and helicopter-pilot Andrew Hefford for their valuable assistance in the field. Additionally, we acknowledge John Southward for his technical support. Ice-penetrating radar data used in this study were acquired by NASA's Operation IceBridge. TerraSAR-X data were provided by the German Aerospace Agency (DLR) under project number HYD1421. We acknowledge support by the High Performance and Cloud Computing Group at the Zentrum für Datenverarbeitung of the University of Tübingen, the state of Baden-Württemberg through bwHPC and the German Research Foundation (DFG) through grant no INST 37/935-1 FUGG. We thank Jan De Rydt for editing and two anonymous referees for their insightful feedback.

#### **Author contributions**

CTW conceived the study, led data analysis and drafted the manuscript. RD participated in fieldwork and contributed to writing. NN processed Sentinel-1 interferograms. JL, HH, SK and WSL facilitated fieldwork from Jang Bogo Station. OJM processed TerraSAR-X data. VH processed GPS data. All authors discussed results, implications, provided feedback and approved of the final manuscript.

#### Data availability

Model outputs are available from CTW. Sentinel-1 imagery is available from the Copernicus Open Access Hub (https://scihub.copernicus.eu/). Sentinel-1 interferograms are available from NN. TerraSAR-X interferograms are available from OJM. GPS data are available at https://doi.pangaea.de/10.1594/PANGAEA.936090. All other data sources mentioned in this study are detailed within the text.

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
