# Peer review of "Monitoring Shear-Zone Weakening in East Antarctic Outlet Glaciers through Differential InSAR Measurements"

_EGUsphere, 2024_

## Author Comment (AC1)

We sincerely thank Reviewer 1 for their insightful feedback. We agree that both major points—the reference value and the physical origin of shear zone weakening—are valid. In response, we have: (1) conducted additional model simulations demonstrating that the heterogeneous signal can also be reproduced by lowering the homogeneous Young's modulus from 1.6 GPa to 1.0 GPa; (2) substantially revised the discussion section to better contextualize these findings; and (3) rephrased sections of the manuscript that previously implied resolution of anisotropy from satellites. We appreciate the constructive comments and provide detailed responses below.

RV1-1:

This paper presents a model/data comparison looking at ice-shelf flexure at Priestley Glacier in Antarctica. The authors use a series of differential interferograms to estimate tidal displacement and examine the spatial pattern of these displacements. The time series is validated with GPS from 2018. They compare the observed displacements with three models of elastic flexure, which vary in terms of their assumptions about the geometry of the problem (ice thickness) and about the Young's modulus. From this comparison, they argue that the shear margins are extremely weak (20% of the expected strength). They conclude that DInSAR can be used to understand shear-margin strength.

This summary accurately captures the main findings of our study, and we appreciate the reviewer's thorough engagement with our work. You correctly point out  some aspects in our methods that were not as clearly conveyed as intended. In the revised manuscript, we restructured key sections, particularly regarding the derivation of synthesized tidal displacements from DInSAR and how they inform our conclusions about shear-margin weakening. We hope this will improve the reproducibility of our approach and address any remaining uncertainties.

RV1-2:

The paper is likely to be of interest to the readership of *The Cryosphere*. It is reasonably well written, though I need some clarification at a few points, and the figures are clear and support the narrative (though some small changes are needed for legibility). However, I am skeptical of the conclusions, as described in "major issues." Specifically, I think the central claims in the abstract are not supported by the results, and it is unclear to me how the results will hold up to more careful checks on their robustness. I also have some reservations about the claims about the physical origins of this signal, assuming it is robust —I do not think fabric is the likely cause. If the authors can demonstrate that the signal of shear-margin weakening is not a byproduct of their assumptions but rather a robust feature of the data, and the fabric-related conclusions were either better supported or removed, I would be supportive of seeing this work in *The Cryosphere*.

We understand that the major issue is linked to our validation with two GPS stations located along the central flowline. Here, you correctly point out that the local heterogeneous model does not improve the fit. In fact, at these specific locations, all model scenarios provide similar predictions showing that these areas are comparatively insensitive to our modeling choices. This was not clearly communicated and hence correctly triggered your concern. However, as we move away from the GPS point locations and into the entire flexure zone (Fig. 11, Fig. 9f-h) the differences are clearer and also the

better fit of the local heterogeneous model becomes apparent. We have changed relevant sections in the revisions to make this more clear.

We also agree that out study does not provide conclusive evidence what mechanism may cause the  shear-zone weakening. While other studies have provided evidence for the role of preferred ice-fabric patterns in shear zones, our data do not substantiate these claims and other mechanisms are also possible. We have weakened our interpretations in this regards both in the abstract and the discussion. Also as a response to RV2-4, we now include more sensitivity experiments improving the robustness of our conclusions.

Major issues

RV1-3:

Unless I have missed something essential, the central claims of the paper as presented in the abstract are not supported by the results. The abstract claims "we find that a five-fold reduction of the Young's modulus in the shear zone, i.e. an effective shear-zone weakening, reduces the root-mean-square-error of predicted and observed vertical displacement by 84 %, from 0.182 m to 0.03 m." However, it seems this number is derived by comparing the unmodulated tidal height to the observed displacement, which fails to account for any elasticity. From line 386, it is clear that the 0.182 is the unscaled tidal forcing. In fact, the local heterogeneous model *underperforms* the local homogeneous one (0.029 vs 0.027 m misfit), so it would be more appropriate in the abstract to say that shear-zone weakening is unable to improve the fit! The central claim of the paper instead needs to rest on the comparison between the local homogeneous (or perhaps control) model and the heterogeneous model—this is what tells us the effect of the shear-zone weakening. All that can really be said based on the misfit is that an elastic model is useful—nothing about the three different models is conclusive with regards to misfit, as acknowledged by the authors at line 333. The matching claim in the conclusions ("we demonstrated that reducing ice stiffness in lateral shear zones significantly improves the accuracy of vertical displacement predictions, particularly along the grounding zone," line 477) is also unsupported by the results.

The reviewer is correct that the analysis based solely on the GPS point measurements does not provide significant evidence for either model scenario (related RV1-2). Our conclusion that a reduction in Young's modulus, representing effective shear-zone weakening, improves the fit between model and observations compared to a homogeneous case is based on an area-wide comparison, as shown in revised Fig. 11.

To clarify this in the revised manuscript, we explicitly delineated this area in revised Figs. 9f/g/h and rephrased the corresponding parts of the abstract and main text. The GPS site primarily serves to demonstrate that scaling tide model output with a measured (or modeled) alpha value is generally useful and that a 1.3-hour viscoelastic delay appears to be present at this site. However, this delay reduces the RMSE between the tide model and GPS data by only 2 mm (Fig. 12b), making it negligible within the context of this study and further supporting the elastic approximation. See new section 5.4 in this context.

RV1-4:

The problem with the misrepresentation above is that it forces the authors to wade into a more complicated comparison in terms of alpha. More physical explanation of alpha is warranted, as discussed in the general comments below, but in terms of the effect on results doing the comparison in terms of alpha obscures the effects of uncertainty and error. We need a careful analysis of these uncertainties and errors to understand if the comparison in terms of alpha is in fact meaningful—as is, I find the error analysis in the paper insufficient. I do not think a single, fixed value of E to treat as a reference that easily justified (for example, the 2019 paper by the same authors has 1.0±0.56 in the abstract). I do not see how the authors can exclude the possibility that there is substantial variation in E because of things like temperature, and that the mean value is incorrect, which in combination may explain much of the misfit. Similarly, the conclusions of the paper rest on the better fit of the model with weakened Young's modulus in the shear margins, but there is not systematic evaluation of how changes in Young's modulus affect the misfit. Maybe reducing the value elsewhere would produce a better fit—we simply do not know. Without a more systematic comparison, and without a clear evaluation of how uncertainty in the parameters assumed constant affect the results, I am not confident that we can in fact conclude that the authors robustly detect a signal in the shear margins.

We agree that E can vary for a number of reasons (temperature being one of them) that were not prominently mentioned enough in the first draft. To clarify, here we use  E = 1.6 GPa as background values as derived in *Wild et al., 2017* from tiltmeter data from the McMurdo Ice Shelf and further applied in *Wild et al., 2018*. The E=1.0 GPa emerged from the Darwin Glacier (Wild et al., 2019), which supports your statement that values of E in situ are far from constant/known (also pointed out by RV2-5). Moreover, the stiffness scales with the ice thickness cubed (Eq. 4) illustrating that the model geometry is important. In order to account for this comment (and also in line with RV2-4) we conducted additional simulations with the synthetic model setup (Fig. A2) and the local homogeneous model and now investigate reducing the homogeneous Young's modulus (Fig. 12 and new section 5.1).

[Figure]

These additional model experiments show that reducing the homogeneous Young's modulus from E=1.6 GPa to E=1.0 GPa does indeed produce a similar result as our best fit heterogeneous experiment. As a consequence, we re-designed our discussion section.

RV1-5:

I also do not buy the argument that fabric is likely to explain these observations. The authors conflate the viscous anisotropy of ice, which is very strong (an order of magnitude weakening or hardening) and anisotropy of the elastic properties, which are much weaker. There has been extensive work on this topic in the seismic literature, so we have a reasonable number of measurements of the effect of fabric on seismic wave speed, which have found values in the range of 5% and below (e.g., Lutz et al., 2022 https://doi.org/10.5194/tc-16-3313-2022, Rathmann et al., 2022, https://doi.org/10.1098/rspa.2022.0574, and many references therein). Since seismic waves are elastic, I would expect the effect of fabric on ice shelf flexure to be similar to its effect on seismic waves, i.e., about 5% rather than the 80% needed to explain the results here. As a starting point, I suggest looking at Rathmann et al., 2022, since they formulate the effects on elastic anisotropy in terms of the anisotropy in the Lamé parameters, which could relatively easily be converted to anisotropy in the Young's modulus and Poisson ratio, and it appears that the value would be on the order of a few percent. Thus, I think section 5.2 should be reworked to acknowledge the limited effect of fabric on elasticity, and

to propose alternatives. Most obvious to me are things like thickness errors, damage, and errors in the value of $E$ used as a baseline. Alternatively, if the authors think this is really a viscous effect, then the validity of the purely elastic model is called into question. The conclusions should be changed to reflect this viscous/elastic difference. I am not convinced that there are grand implications for ice-stream initiation and would certainly need to see more discussion in section 5 if this were to remain in the conclusions.

As already pointed out in responses to RV1-2 and RV1-5 (and RV2-5) we believe that you capture a valid point here and we are thankful for your suggestions. We acknowledge that ice-anisotropy may be a candidate for shear-margin weakening, but how this would imprint on our model assumptions in terms of $E$ is a different story. We have significantly reworked section 5.2 taking your suggestions into account and removed the statements about ice-stream initiation from the conclusions.

General comments

RV1-6:

The least-squares adjustment needs more explanation. It is a bit strange to do least squares with an underdetermined system—I guess this amounts to trying to adjust the tide model as little as possible? What this assumption implies deserves explanation. However, I am confused as to how the misfit is not reduced to zero when the system is underdetermined—is this system of equations not linearly independent? A sentence explaining why there is any residual misfit would help clarify.

Yes, our approach aims to adjust the tide-model output as little as possible to match our DInSAR observations. This assumption implies: (a) that DInSAR observations provide the absolute reference for tidal displacement on the freely floating part of the ice shelf, and (b) that tide-model uncertainties are more significant in the amplitude of tidal constituents rather than their phase.

We further quantify the role of phase uncertainty in Fig. 12, where we apply a net phase shift of 0.37 h to the tide-model output to maximize the match with our GPS data. However, this phase adjustment only improves the RMSE by a few centimeters, suggesting that amplitude discrepancies remain the dominant source of misfit.

The reviewer is correct that the residual misfit is essentially zero (within the computer precision). Reworded.

RV1-7:

I am a bit unclear how the load tide is handled. Is the bed underneath the grounded portion of the glacier truly assumed fixed, so that w=0 there? Or is the load tide assumed to apply only where there is ocean water, neglecting the elastic effect on land upstream? This choice should be clarified and justified in the text.

Yes, in our model, the load tide is only applied on the floating portion of the domain and is not transmitted through the bed to the grounded ice. This choice is based on two key considerations: (a) Magnitude of the load tide: As shown in Fig. 4a/b, the load tide is an order of magnitude smaller than both ocean tides and the inverse barometer effect (IBE). (b) Minimal tidal forcing at the grounding line: The elastic response of the bed to ocean

tidal loading is negligible compared to the displacement observed in freely floating areas. This justifies our assumption that the bed underneath the grounded ice is effectively fixed, meaning w = 0 there.

RV1-8:

The mixture of alpha and w is confusing to me. Line 204 says that alpha is "the mean vertical displacement that can be expected during SAR data acquisition", but based on units it is the fraction of maximum displacement expected. A clear, physically motivated definition of alpha, with units, would help if placed in 3.1.4. Also, we need a bit more physical explanation about how alpha is determined—in particular, I am not clear on what assumptions about spatial variations are employed here.

We acknowledge that this sentence was not correctly phrased. An alpha value of 10% means that a 1 m tidal forcing results in a 10 cm vertical displacement at that location. The "mean vertical displacement" phrasing refers to the fact that alpha is derived by averaging all available interferograms, capturing the expected response over multiple tidal cycles. We now clarify this in Section 3.1.4 and provide a more precise, physically motivated definition of alpha. Alpha is percentage tidal displacement and therefore has no unit.

RV1-9:

Some reorganization of methods and results is needed. Section 4.2 is a confusing mix of methods and results. I am not clear on what these mosaics in Figure 7 are. I am assuming they are DInSAR images aggregated in some way, but it is not clear how. It seems to me that this relates to section 3.1.4, but I am not completely clear. Lines 321 to 326 are methods and so belong under the top-level header of 3.3.3 (this would have helped me understand the motivation of multiple models better there, too).

We appreciate this suggestion and have restructured the revised manuscript accordingly (also in line with RV2-3). The mosaics refer to synthesized DInSAR images, which are based on an alpha-map derived from the original DInSAR images and least-squares adjustment. Figure 7 illustrates the same procedure as Figure 5 but applied across the entire grounding zone rather than just a single point on the freely floating ice shelf.

RV1-10:

I would like to see a brief analysis of how thickness errors would affect the results. I assume this is minor, based on how thickness enters Eq 4, but it would be nice to exclude this completely.

Uncertainties in ice thickness are effectively captured by the difference between the Control Model and the Local Homogeneous Model, which represent the lower and upper bounds of the present ice thickness distribution. The range of flexure curves resulting from these two thickness distributions is smaller than the residual misfit to the observed flexure profile from DInSAR (Fig. 9k). This suggests that ice thickness variations alone cannot explain the observed flexure, and an additional weakening mechanism is required to match DInSAR. In our analysis, weakening of the lateral shear margins provided the best match to the observed flexure profile (Fig. 9l).

Line comments

all agreed

Figure 2: The scales appear distorted in b (Antarctica is the wrong shape). The axes should be checked so that squares are square.

The map inset was indeed scaled, which lead to the distortion.

L209: It is not clear whether the adjusted maps are alpha itself or the DInSAR measurement after adjustment

L230: How does a fulcrum facilitate transmission?

L289: Reduced accuracy makes it sound worse; improvement like this is normally referred to as greater accuracy

The reviewer was right that the least-squares adjustment improves the misfit to virtually zero, so we re-worded this sentence accordingly.

L299: "Notoriously" is hyperbolic and unnecessary. Simply remove it.

Removed.

L305: Not clear what it means to "perform…combination"

DInSAR imagery captures tidal flexure as a combination of tides (+1/-2/+1). Our methodology separates these images into single-tide components, which are then recombined for comparison with the original DInSAR imagery. We refer to the original imagery as 'Measurement' and the recombined version as 'Mosaic.' This step is crucial for demonstrating the robustness of our method, particularly for different DInSAR images acquired at various stages of the tidal cycle. This is why we present Fig. 7, which shows that their difference is essentially zero.

Figure 7: Plotting in blue on top of an image with surface meltwater is just confusing. I suggest making the background image black and white throughout

We have applied a grayscale version of the background image for all figures that use a blue colormap. However, we retained the original colored image for other figures, as the distinct blue ice surface of Priestley Glacier is an important feature to sidestep the firn problem (as also noted by RV2-1).

L380: I think this sentence needs rephrasing—does the IBE really do the reduction?

Reworded. The reduction is from tidal loading AND the IBE

L391: Tide deflection ratio is not defined—is this alpha?

Yes it is.

L429: The crystal lattice typically refers to sub-grain structure (i.e., the arrangement of molecules), not the aggregate of grains as used here. Suggest "the polycrystal" instead.

Reworded.

Throughout: hyphens are only used between double nouns when they modify something. So "raise sea level" is correct, as is "sea-level rise," but it is incorrect to write "raise sea-level." There are a number of errors in this vein in the manuscript.

Re-hyphened accordingly

---

## Author Comment (AC2)

We greatly appreciate the insightful feedback of Reviewer 2. We agree that our analysis does not support ice anisotropy as the cause of (elastic) shear zone weakening and have removed this claim accordingly, for the reasons you outlined. The discussion has been revised to reflect this. We also incorporated additional sensitivity tests on bed stiffness and improved the methods section. We hope the updated roadmap and new Section A2 address your concerns effectively.

RV2-1:

This is a really interesting paper doing some innovative things about investigating the spatial variations in effective Young's modulus near the grounding line that is deeply needed for updating our knowledge about the in situ rheology of ice and tidal processes in the grounding zone. It's a good site to pick for this analysis to sidestep the firn problem and the grounding line migration. It is excellent technical work at the difficult intersection of models and observations.

Thank you for your positive feedback and for recognizing the significance of our work. We deeply appreciate your acknowledgment of the challenges involved in bridging models and observations in the flexure zone.

RV2-2:

Overall the figures are rich and detailed and could generally be accompanied by a bit more narrative explication. I really appreciate figure 4d, I've rarely seen the tidal corrections overlaid like that and it's but helpful and interesting to look at the relative magnitudes and phases. I think figure 9 i-l tells a good story and is helpful to understanding the method.

We incorporated additional narrative explanations where needed to enhance clarity and guide the reader through the key insights presented in the figures.

RV2-3:

Overall, I think the narrative and "thesis" of the paper, and the sequence of what exactly was done, could be made clearer to the reader. Providing a roadmap in the methods section might help. I was surprised when certain aspects of the methods came up, for instance the radar thickness data. For this paper, I would also not assume the reader is at all familiar with alpha maps and elect not to force them to read the preceding paper. A standalone explanation is needed and they can be referred to the previous paper for details.

We reorganized the manuscript for better clarity and flow. We now provide a more detailed, standalone explanation of alpha maps at the beginning of Section 3. Regarding the radar ice thickness data, we decided not to place too much emphasis on these measurements, as they were acquired in November 2013 and are relatively outdated compared to the other datasets used in the study.

RV2-4:

I also think that some of the modeling choices could be fleshed out in greater detail, particularly the value for the Young's modulus of ice, and the assumption of an elastic bed with the spring constant used in the work. I know Sayag and Worster (2013) have a value for the spring constant in there and I'm curious how it compares. Sketching some

uncertainty bounds around these parameters would strengthen the argument that the signal in the flexure can be distinguished enough from the noise to attribute it to the Young's modulus of the ice.

Thank you for this insightful comment, and we appreciate the reviewer bringing it up. We performed additional experiments using our idealized model setup to better understand the sensitivity of ice-shelf flexure to variations in bed stiffness. The results from this experiment indicate that up to 2-3% of the 10% change in flexure might be influenced by the bed stiffness. These sensitivity experiments have been added to the Appendix.

[Figure]

A direct quantitative comparison with Sayag and Worster (2013) is challenging, as their model experiments use stiff-fixed and soft-free grounding line boundary conditions, while our fulcrum is soft-fixed. The main takeaway from Sayag and Worster's work is that the discrepancy between laboratory-derived Young's modulus and in situ-derived Young's modulus could be explained in their model by altering the boundary condition from stiff-fixed to soft-free, which then allows the grounding line to migrate inland during high tide. However, we circumvent this effect by choosing the Priestley Glacier study site, where tidal grounding line migration is minimal due to the steep bed slopes present.

We clarified these modeling choices and incorporate a discussion of the sensitivity of the flexure signal to variations in bed stiffness, as well as consider uncertainty bounds around the parameters used, to strengthen the argument that the signal in the flexure can indeed be attributed to variations in the Young's modulus of the ice.

RV2-5:

I also think the connection to fabric is somewhat tenuous and might be rebuilt somewhat around the surprising idea that we just don't know very well what affects Young's modulus in situ. It may be worth touching some of the older and newer literature around laboratory experiments on the stiffness of ice. I might restructure the discussion to include a general discussion of the main takeaways in the results before getting into the weeds of viscoelasticity that hasn't been brought up yet (though I understand the need for the justification of the elastic model, which I am fine with). There are some things I found interesting that didn't get returned to- the shape of the bulge in figure 1e for instance. Overall this makes a clear contribution to the field and will be made even better with a clearer and more self-contained explanation of the methods.

We agree with both reviewers that our analysis does not provide evidence that ice anisotropy is the cause of the observed softening in the shear zones. We clarified this in the revised manuscript and focus more on the uncertainty around in situ Young's modulus.

We also incorporated relevant literature on ice stiffness and restructure the discussion to highlight key results before addressing viscoelasticity.

The shape of the buldge depends on the amount of shear-zone weakening. We added the buldge of the best heterogeneous model (20% E) to a new Figure. For the sake of completeness, here are the other buldges. The buldge disappears for Young's modulus

[Figure]

>60%.

131: Why 1.6? There's a range found in other studies (mostly not referenced here).

1.6 GPa is the value of the Young's modulus that was derived by Wild et al., 2017 and further employed in Wild et al., 2018 from the McMurdo Ice Shelf. We think it's a better reference value for a homogeneous case than 1.0 GPa as derived from the more complicated Darwin Glacier in Wild et al., 2019, because the reduced value might be a similar signal of shear zone weakening at Darwin Glacier (which would be an interesting follow-up study, to proof that a similar flexure pattern can be produced with E=1.6 GPa and weakening Darwin Glacier's shear zones)

146: I'm curious about what only means here, was there consideration given to changing the boundary conditions?

"Only" refers to model experiments by Still et al. (2022), where they suggested that either the sidewall boundary condition or the assumption of a uniform Young's modulus could be

incorrect. Their GNSS observations indicated that a heterogeneous Young's modulus or partial contact with the sidewall improved model fit. We also tested changes to the grounding line boundary condition. Although our model simulations are in planar view (so partial ungrounding on sidewalls isn't supported), we experimented with clamped versus fulcrum boundary conditions. We found that a fulcrum more realistically produces the s-shaped flexural signature, while the clamped condition results in an "onion-like" structure.

[Figure]

160: what does "adjustments for the input data used for predictions" mean?

It means that tide-model output has to be 'adjusted' to agree with DInSAR measurements. We added a roadmap of the method at the beginning of Sect. 3

185: this is a bit confusing and could use a more extended discussion in plain language, connecting back to the method at large.

This step is essential for aligning our model predictions with the DInSAR observations. If we use the tide model output without adjustment, the mismatch with DInSAR can be as large as ±15 cm, or several fringes, making direct comparison meaningless. To address this, we calibrate the tide model to match observed displacements in freely floating ice, ensuring consistency between model input and satellite data. We hope this adjustment is now clearly outlined in the roadmap and revised Methods section.

200: From this description I don't know what an alpha map is, and certainly couldn't reproduce it from this paper. Even though referencing another paper, a recapitulation here would help, especially as it seems important to the methods of this paper in discerning what we're comparing.

We reworded Sect. 3.1.2 "Derivation of an alpha-map from DInSAR measurements".

215: what does "processed the data" mean?

It means how we converted the raw data from the TRIMBLE receivers into coordinates.

221: sources for k = 5 MPa? My impression is this is very uncertain and probably varies by a lot.

The value of k = 5 MPa comes from Walker et al. (2013), which was based on observations of 1 cm uplift reported by Heinert and Riedel (2007) on the Ekström Ice Shelf. We appreciate the reviewer's note on the uncertainty of this value, and we hope that our sensitivity experiments (RV2-4) and the section in the appendix (A2) help address this concern.

226: first mention of effective Young's modulus? What are you defining it as in this paper?

We use the term "effective" Young's modulus to acknowledge that, in natural, real-world conditions, ice behaves differently from theoretical or laboratory-based conditions due to factors like ice fabric, temperature variations, and impurities. Added this sentence to the revised manuscript.

245: what was the magnitude of these corrections?

A +1 hPa anomaly of atmospheric pressure translates to an instantaneous −1 cm contribution to the tidal forcing, as shown by Padman et al. (2003). Added this sentence to the revised manuscript.

258: I could use another sentence or two on what thickness update means here.

The thickness update refers to the difference in ice thickness between the Control Model, which uses the BedMachine dataset, and the Local Model, where we perform our own inversion of the ice-shelf freeboard to estimate the ice thickness. Added a short statement.

269: I think this is the first mention of radar thickness data, how do you use it?

We use the radar ice thickness transect primarily to compare it with the BedMachine dataset and our 'thickness update' from the inversion of freeboard. However, we do not focus heavily on this comparison as the IceBridge radar thickness data is from 2013, which is relatively outdated. The IceBridge flight path, however, follows a flowline and is therefore where we define our profile through alpha-maps and model solutions presented in later Figures.

288: it's not abundantly clear to me what least squares adjustment means here.

As also noted by RV1, the approach involves adjusting the tide model output at t1,t2,t3 as little as possible to match the DInSAR observations on the freely-floating part of the ice shelf. The challenge is that DInSAR represents a double-differential tide from three individual time points:

(t1−t2)−(t2−t3).

The least squares adjustment finds the smallest offsets Δx,Δy,Δz to correct the model output at t1,t2,t3 while best matching the observed tide. This is done by minimizing the following expression:

min∑[((x+t1)−(y+t2))−((y+t2)−(z+t3))−observed value],

where x,y,z represent the adjustments to the model outputs at times t1,t2,t3, respectively. This is essentially done for all 31 DInSAR measurements at once by solving the combination matrix.

341: why might they over then underestimate? If you discuss later, you can point the reader there. Also, we may want to be careful about prescribing the exact location of the grounding line from interferograms. The extent of upstream flexure is likely past the grounding line because ice is thick.

At ~16 km downstream of the grounding line, the lateral grounding line on the Nansen Ice Shelf may be incorrect due to noise in the interferograms, which prevented precise delineation. This coincides with the true right shear zone, which runs parallel to the

Nansen's grounding line. For our model, we interpolated between visible fringes, so this might be an artifact.

Your second point is interesting. We agree that individual interferograms often capture a hinge line rather than the true grounding line, as shown in previous studies. However, at Priestley Glacier, we compared a grounding line derived from an alpha map (based on 31 interferograms across tidal cycles) with a TerraSAR-X-derived grounding line and found a good match. Alpha maps average out spurious displacement due to beam flexure (Rack et al., 2017), making them less susceptible to such biases.

356: how these statements follow from one another is unclear to me.

Thank you for pointing this out. Up to this point, we have compared model solutions along a single transect (the IceBridge flight path). Here, we extend the model-observation comparison to the entire grounding zone. To avoid confusion, we outlined this region of interest (ROI), which is also used for Fig. 10, in the corresponding figure panels 9f/g/h.

Minor comments:

75: sentence fragment

Reworded

288: "millimeters"

Reworded

---

## Author Response (AR2)

I would like to thank the authors for their extensive work to address my comments (I was reviewer 1 during the first round of review). All my major concerns have been addressed. I find the manuscript much more convincing now, and I think it is better presented as well. It is nice work, and I'd like to see it published. I have only minor comments on the new version (line numbers refer to the track-changes pdf).

We thank the reviewer for taking the time to carefully read the revised manuscript and for the positive and constructive feedback throughout the review process. We agree with all remaining minor comments. The main changes made in this revision include: (1) clarifying early in the Methods section how the GNSS data are integrated into the analysis, and (2) explicitly addressing the non-uniqueness of our model solutions in the Conclusion.

(line numbers in our responses refer to the manuscript.pdf)

Section 3 (L160-175): Foreshadowing the role of the GNSS in the analysis would help the reader

Agreed. We now clarify that the GNSS data are used to validate both the homogeneous and heterogeneous model experiments. We find that incorporating any alpha-map significantly improves the fit between observed and modeled displacements. However, the GPS sites themselves are not particularly sensitive to the specific modeling choices, which motivates a broader evaluation across the entire tidal-flexure zone. Finally, we note that the GPS records are also used to estimate the timing of tidal oscillations at the two sites to assess the validity of the elastic approximation.

Fig 2: It would be nice to have a value on the bottom of the colorbar Agreed. The colorbar starts from 1 m/yr. Added this value to Fig. 2b.

L427: No comma after means, add comma after choices Agreed and changed accordingly in L414-415.

L437: percentage of Agreed and added 'of' to L425

L614: The homogenous model with E=1.0 should be mentioned as an alternative, equally valid explanation

Agreed and added a sentence to L563-566 about the two scenarios.